# Refining the *Camelus dromedarius* Myostatin Gene Polymorphism through Worldwide Whole-Genome Sequencing

**DOI:** 10.3390/ani12162068

**Published:** 2022-08-14

**Authors:** Silvia Bruno, Vincenzo Landi, Gabriele Senczuk, Samantha Ann Brooks, Faisal Almathen, Bernard Faye, Suheil Semir Bechir Gaouar, Mohammed Piro, Kwan Suk Kim, Xavier David, André Eggen, Pamela Burger, Elena Ciani

**Affiliations:** 1Department of Biosciences, Biotechnologies and Biopharmaceutics, University of Bari “Aldo Moro”, 70126 Bari, Italy; 2Department of Veterinary Medicine, University of Bari “Aldo Moro”, Valenzano, 70010 Bari, Italy; 3Department of Agricultural, Environmental and Food Sciences, University of Molise, 86100 Campobasso, Italy; 4Department of Animal Sciences, University of Florida, Gainesville, FL 32610, USA; 5Department of Public Health, College of Veterinary Medicine, King Faisal University, Al-Ahsa 31982, Saudi Arabia; 6Camel Research Center, King Faisal University, Al-Ahsa 31982, Saudi Arabia; 7CIRAD-ES, UMR SELMET, 34398 Montpellier, France; 8Department of Biology, Abou Bakr Belkaid University of Tlemcen, Tlemcen 13000, Algeria; 9Department of Medicine, Surgery and Reproduction, Institut Agronomique et Vétérinaire Hassan II, Rabat BP 6202, Morocco; 10Department of Animal Sciences, Chungbuk National University, Chungbuk 28644, Korea; 11Illumina, Agrigenomics, 91000 Evry, France; 12Research Institute of Wildlife Ecology, Vetmeduni, 1160 Vienna, Austria

**Keywords:** myostatin, dromedary, single nucleotide polymorphisms, indels

## Abstract

**Simple Summary:**

The dromedary is a multipurpose livestock species. Growing interest in dromedary meat production, increasing intensification of meat production systems and rising business turnover for the dromedary athletic competition sector point to the need for a more effective genetic selection. To this aim, the identification of genetic markers, to be implemented in phenotype–genotype association studies, represents a pre-requisite. The myostatin gene is known to affect, to a various extent, muscularity in several animal species. In this study, we investigated the dromedary myostatin gene sequence variation through the analysis of data from 183 worldwide animals. A total of 99 variants were detected in the target region. Through a bioinformatic approach, we explored the possible functional effects of the detected variants. Several hints emerged, suggesting that natural variants at the dromedary myostatin locus may determine quali-quantitative changes in the myostatin expression, thus likely impacting on dromedary muscularity. Further efforts are needed to collect reliable phenotypic data on dromedary muscularity and racing performances, thus allowing for the unveiling of those genetic markers that are associated with the phenotypic variability in muscularity-related traits. Under intensive rearing systems, large-scale screening of genotypes at those markers in dromedary populations will effectively orient selection decisions and positively impact on rates of genetic gain.

**Abstract:**

Myostatin (*MSTN*) is a highly conserved negative regulator of skeletal muscle in mammals. Inactivating mutations results in a hyper-muscularity phenotype known as “double muscling” in several livestock and model species. In *Camelus dromedarius*, the gene structure organization and the sequence polymorphisms have been previously investigated, using Sanger and Next-Generation Sequencing technologies on a limited number of animals. Here, we carried out a follow-up study with the aim to further expand our knowledge about the sequence polymorphisms at the myostatin locus, through the whole-genome sequencing data of 183 samples representative of the geographical distribution range for this species. We focused our polymorphism analysis on the ±5 kb upstream and downstream region of the *MSTN* gene. A total of 99 variants (77 Single Nucleotide Polymorphisms and 22 indels) were observed. These were mainly located in intergenic and intronic regions, with only six synonymous Single Nucleotide Polymorphisms in exons. A sequence comparative analysis among the three species within the *Camelus* genus confirmed the expected higher genetic distance of *C. dromedarius* from the wild and domestic two-humped camels compared to the genetic distance between *C. bactrianus* and *C. ferus.* In silico functional prediction highlighted: (i) 213 differential putative transcription factor-binding sites, out of which 41 relative to transcription factors, with known literature evidence supporting their involvement in muscle metabolism and/or muscle development; and (ii) a number of variants potentially disrupting the canonical *MSTN* splicing elements, out of which two are discussed here for their potential ability to generate a prematurely truncated (inactive) form of the protein. The distribution of the considered variants in the studied cohort is discussed in light of the peculiar evolutionary history of this species and the hypothesis that extremely high muscularity, associated with a homozygous condition for mutated (inactivating) alleles at the myostatin locus, may represent, in arid desert conditions, a clear metabolic disadvantage, emphasizing the thermoregulatory and water availability challenges typical of these habitats.

## 1. Introduction

Myostatin (*MSTN*), also known as Growth Differentiation Factor-8 (*GDF8*), is a member of the transforming growth factor (*TGF-β*) superfamily. It is a negative regulator of skeletal muscle development and homeostasis in mammals [1,2]. The gene sequence has been highly conserved across vertebrate species throughout evolution [3]. It consists of three exons and two introns. Myostatin is synthesized as a 375 amino-acid precursor protein called pre-myostatin, that undergoes post-translational modifications in order to become biologically active [4].

Inactivation, through C-terminal deletion, of the *MSTN* gene was shown to determine a two- to three-fold increase in the skeletal muscle mass in mutant compared to wild-type mice [5]. The *MSTN*-knocked out rabbits were shown to generate a heritable phenotype with hyperplasia and hypertrophy of the muscle fibers, which could be valuable in improving rabbits’ meat production [6]. MSTN is, indeed, among the most studied genes in meat-producing animals. The mutations at this gene have been shown to naturally occur in several livestock species. In particular, loss-of-function mutations have been associated with a hyper-muscular phenotype, known in cattle as “double muscling” (DBM). The Belgian Blue cattle breed represents a remarkable example of *MSTN*-dependent muscular hypertrophy. In this breed, an 11-bp deletion (g.821-831del11) results in the loss of three amino-acids and a frame-shift mutation, leading to a premature stop codon [7]. The double-muscled cattle display not only an increase in skeletal muscle mass but also a reduction in fat mass [8], with the consequent production of very lean meat. This phenomenon is not surprising, given the fact that *MSTN* has been demonstrated to play key roles not only in myogenesis, but also in adipogenesis [8,9]. In a quantitative trait locus (QTL) survey for growth traits, performed using an F2 Duroc x Pietrain pig resource population, with the Belgian Pietrain breed known for being heavily muscled, Choi et al. [10] identified a locus associated with the *longissimus muscle* area in a region that includes *MSTN*. In the Belgian Texel (Beltex) sheep, Clop et al. [11] reported a mutation creating a potentially illegitimate microRNA target site in the myostatin gene affecting muscularity in sheep. All of the above examples nicely highlight how the Belgian selection of superior-performing animals for extreme muscularity and lean meat percentage has impacted the sequence variability at the *MSTN* locus in these species. Although the double-muscled phenotypes have never been described in the equine species, in thoroughbred horses a SINE insertion in the promoter of the *MSTN* gene inducing a decrease in myostatin expression [12,13] was found to be associated with increased muscle mass [14], thus highlighting that regulatory mutations may more finely tune the hyper-muscularity in livestock species. In addition, the horse SINE also positively affected the aerobic capacity, speed and stamina [15]. A similar phenomenon of increased athletic performance was observed in racing whippet dogs presenting a MSTN mutation leading to a premature stop codon [16].

The dromedary camel (*Camelus dromedarius*) represents an economically relevant livestock species, especially in the countries affected by severe climate change and desertification, assuring the production of milk, meat, leather, wool and a workforce, even in very harsh conditions. The interest in dromedaries and their products is also rapidly growing in Europe where camel farms have been established, such as “La Camélerie” (https://www.lacamelerie.fr/ accessed on 7 August 2022), in France, and the Kamelenmelkerij Smits (https://www.kamelenmelk.nl/ accessed on 7 August 2022), in Netherlands. Moreover, the Emirates Industry for Camel Milk and Products (EICMP), owner of the “Camelicious” brand (https://camelicious.com/ accessed on 7 August 2022), has become the first Middle Eastern company to obtain a license to export dromedary products to the European Union. In recent years, there has been an increase in the demand for camel meat as an alternative lean food, characterized by a lower fat content, lower cholesterol and a higher proportion of polyunsaturated fatty acids than most consumed meat from cattle and sheep [17]. In addition, the dromedary is traditionally exploited, mainly in the Arabian Peninsula countries, for racing and beauty contests. Based on the above, its morphological features of muscularity may influence the economic value of the animal. Nevertheless, despite the fact that differences in muscularity were observed among distinct populations [18] and/or individuals [19], to our knowledge no evident hyper-muscular phenotype has so far been described in dromedaries.

In *C. dromedarius*, the *MSTN* gene-structure organization and the sequence polymorphisms have been previously investigated, using Sanger [20] and Next-Generation Sequencing (NGS) [21] technologies. No report of loss-of-function mutations exists so far in this species, and the above mentioned studies highlighted a lower degree of sequence variation than in other livestock species, possibly related to its peculiar evolutionary history, characterized by: (i) a domestication occurring from a markedly bottlenecked wild genetic stock [22] and (ii) a potentially not particularly strong human-based positive selection pressure on the muscularity traits. Indeed, although camels were largely used as pack animals in antiquity, the obvious attitudinal advantage deriving from higher muscle mass has been attained preferentially through inter-specific hybridization with *C. bactrianus* [23]. Moreover, the adaptive phenotypic and genetic plasticity phenomena typical of mammals living in xeric environments may well have represented negative constraints on muscle mass development in order to cope with thermoregulatory and water availability challenges [24,25]. In order to check whether the low sequence variation previously reported [20,21] in *C. dromedarius* at the *MSTN* locus could also be affected by the relatively limited number of animals enrolled in the studies, we extended the NGS analysis here to a world-wide set of 183 dromedaries. The sequence polymorphisms results are presented, and their possible functional effect is predicted through in silico analysis.

## 2. Materials and Methods

### 2.1. Sampling and Whole-Genome Sequencing

Within the frame of an Illumina^®^-funded project (2019 Agricultural Greater Good Initiative), through a large international collaborative effort, 161 *Camelus dromedarius* biological samples (whole blood and hair follicles) were collected from 18 countries (Appendix A), representative of the geographic distribution range of this species. Whenever possible, the samples were collected to avoid closely unrelated animals, based on herders’ knowledge. The vast majority of the samples were collected for previous research projects (FWF P24706-B25 and P29623-B25, PI: P. Burger; EU Arimnet2 CARAVAN, PI: E. Ciani) during routine veterinary procedures, therefore no ethical assessment and permit was specifically required for sampling (art. 1, comma 5 of the Directive 2010-63-EU). Notwithstanding that, procedures to avoid any possible animal distress during the samples’ collection were put in place, according to local consolidated practices. The hair samples from Ethiopia were specifically collected for this project by the researchers of Ethiopian Biotechnology Institute, with no ethical authorization required for hair sampling (art. 1, comma 5 of the Directive 2010-63-EU). In order to assure local cultures’ sensitivity [26] in the traditional dromedary countries, communication tools based on dialogue as a basis for trust-building and informed (oral) consent were adopted.

The transfer of biological material was completed following the example material transfer agreement provided as the annex to the FAO guideline on the molecular genetic characterization of animal genetic resources (FAO, 2011).

The DNA from whole blood was extracted using the Qiagen DNeasy Blood and Tissue kit (Qiagen, Venlo, The Netherlands). The DNA from the hair follicles was extracted with a DNA salting-out method [27]. The DNA quantity and quality was assessed with a Qubit fluorometer (Invitrogen, Vienna, Austria).

Paired-end library preparation and sequencing steps were carried out by the Illumina Solutions Center in San Diego, CA. In brief, the DNA was first quantified using the Quant-iT PicoGreen dsDNA Assay Kit (Thermo Fisher Scientific, Waltham, MA, USA). Between 100–500 ng of DNA was used as the input to prepare the libraries, using the Illumina DNA Prep library kit and the IDT for Illumina DNA/RNA UD Index set (Illumina, San Diego, CA, USA), following the reference guide. The final libraries were quantified using the Quant-iT PicoGreen dsDNA Assay Kit and normalized to 2 nM using an average region size of 600 bps. The libraries with unique barcodes were pooled together and sequenced on the Illumina NovaSeq 6000 system using the NovaSeq 6000 S4 300 Cycle Reagent Kits (Illumina, San Diego, CA, USA). In addition to the above, we also capitalized the publicly available whole-genome sequence data for this species (Appendix A), by retrieving 22 sequence sets representative of nine countries.

### 2.2. Mapping and Variant Calling

The raw reads from the 183 considered animals were analyzed by using the DRAGEN Germline App v.3.9.5 on the BaseSpace™ platform (Illumina, San Diego, CA, USA), using the default setting parameters. The DRAGEN workflow includes both alignment and variant calling algorithms: the reads were first mapped and aligned to the *C. dromedarius* CamDro3 reference genome (Genebank accession: GCA_000803125.3), then the variant calling was performed producing a genotype vcf.gz output file overall, including 17,679,716 variants.

### 2.3. Characterization of Variants in the Myostatin Locus

The genotype file was loaded into the RStudio software v.1.4.1103 (R version 4.0.3) and the variants in the range of ±5 kb upstream and downstream of the myostatin gene were retained by using the Tidyverse package v.1.3.2. Overall, our target region was 16,757 bps long, including 6757 bps of the *MSTN* gene and 5000 bps upstream and downstream (chr5: 58,454,553–58,471,309). The variants were classified based on their nature (SNPs or indels), their localization (5′-UTR, exonic, intronic, 3′-UTR, intergenic) and their occurrence in the previously published paper by Favia et al. [21]. In addition, the popular [28] Tajima’s D neutrality test [29], as implemented in VCFtools v0.1.13 [30], was carried out on the chromosome (5) harboring the *MSTN* locus, performing the analysis on 2000 bps windows. If the D values are too large or too small, the neutral null hypothesis is rejected. As a rule of thumb, we considered here that values greater than +2 (balancing selection) or less than −2 (directional selection) are likely to be significant [28]. For the SNPs detected in the exonic regions, the effect of the polymorphisms on the *MSTN* amino-acid sequence and the codon usage frequency patterns were evaluated. The latter were assessed via the ATGme software [31], using the *C. dromedarius* codon usage table available at http://www.kazusa.or.jp/codon/cgi-bin/showcodon.cgi?species=9838 (accessed on 7 August 2022).

### 2.4. Inter-Specific Comparative Sequence Variant Analysis

A sequence comparative analysis of the target myostatin region described above was carried out by BLASTing, the publicly available *Camelus dromedarius* myostatin reverse-complement sequence (GenBank accession: GCA_000803125.3) against *Camelus bactrianus* (GenBank accession: GCA_000767855.1) and *Camelus ferus* (GenBank accession: GCA_009834535.1) corresponding sequences. We opted here for using the reverse-complement sequence of the *C. dromedarius* CamDro3 reference genome (Genebank accession: GCA_000803125.3) in order to allow for an easier comparison with the previously published paper by Favia et al. [21]. Hence, all of the subsequent analyses were also performed on the CamDro3 reverse-complement sequence.

### 2.5. In Silico Functional Prediction

The web-based software TFBIND [32], available at https://tfbind.hgc.jp/ (accessed on 7 August 2022), was used to identify the transcription factor binding sites (TFBS) and their possible disruption due to the presence of Single Nucleotide Polymorphism or indels. The TFBIND analysis was carried out, considering the 2 kb upstream the *C. dromedarius* myostatin gene. In addition, the online bioinformatic tool Human Splicing Finder (HSF) [33], available at https://www.genomnis.com/ (accessed on 7 August 2022), was used to perform a Sequence Analysis (Version 1.5.1.). The HSF tool employs several matrices to predict the variants’ putative effect on the splicing motifs, including the acceptor and donor splice sites, the branch point and auxiliary sequences known to either enhance or repress the splicing. Both of the above analyses were carried out, for each variant, using the two input sequences harboring the alternative alleles.

## 3. Results and Discussion

### 3.1. Sequence Analysis

Our sequence variant analysis performed in the ±5 kb upstream and downstream region of the *MSTN* gene highlighted the presence of a total of 99 variants (Appendix A).

Out of them, 77 were Single Nucleotide Polymorphisms (SNPs) and 22 were indels. Based on the localization, 72 variants were intergenic (namely, 38 variants in the 5 kb region upstream of the MSTN gene and 34 variants in the 5 kb region downstream of the MSTN gene), 1 SNP in the 5′ UTR region, 1 SNP in exon 1, 11 variants in intron 1, 1 SNP in exon 2, 9 variants in intron 2, and 4 SNPs in exon 3 (Appendix A). The 11 SNPs, previously detected by Favia et al. [21], in the sequence overlapping with our target region using a dataset of nine dromedaries, were also found in our 183 samples’ dataset. Out of the six SNPs that we detected in the exonic regions of the myostatin gene, none of them was responsible for the amino acid substitution at the protein level (Table 1).

The frequency of codon usage was tested, for the six SNPs in Table 1. Two rare codons (CCG and GTA, Allele 2 codons in Appendix A) were detected in the reference *MSTN* sequence in exon 1 (frequency 8.1 ‰) and exon 2 (frequency 5.5 ‰). These were converted into common codons (CCA and GTC, Allele 1 codons in Appendix A) when the alternative alleles detected in our study were fed into the ATGme software. On the other hand, when considering the alternative alleles, a codon in exon 3 (CAT) resulted in being rare (6.4 ‰), while being common when considering the reference sequence (codon CAC).

An average density of one SNP every 217 bps was observed in our 16,757 bps considered region. This value is much higher than that reported by Favia et al. [21], who mention an average density of one SNP every 1.5 kbps. It must be noted that the number of whole-genome sequenced animals differed markedly between the two studies, with nine animals sampled from the Gulf countries (Kingdom of Saudi Arabia, United Arab Emirates, Qatar), Pakistan, Sudan, Kenya and Spain (Canary islands) in Favia et al. [21], and 183 animals representative of 19 countries from three continents, in this study, respectively. Despite the fact that the average SNP density observed in this study is generally comparable with the expected figures in mammals (e.g., in the ovine species, on average one SNP every 204 bps, by re-sequencing the nine animals drawn from different breeds [34]; in the bovine species, on average one SNP every 104 or 434 bps by sequencing the leptin or the amyloid precursor protein genes, respectively, in 22 individuals from the two subspecies *Bos taurus* and *Bos indicus* [35]) and in aves (e.g., in the chicken, about one SNP every 200 bps, by re-sequencing three animals from a corresponding number of domestic chicken breeds contrasted with the sequence of their wild ancestor, the red jungle fowl [36]; in ducks, on average one SNP about every 86 bps, through the genotyping-by-sequencing of 49 animals from the same flock [37]), still, the fact that, to reach such a value, a number of animals much higher than in the above mentioned studies had to be sequenced points out that a generally lower sequence variability exists in *C. dromedarius*, thus confirming our previous results on a more limited numbers of animals [20,21].

A rather clear conservative evolutionary constraint appears to exist on the myostatin locus, as highlighted by the much higher level of polymorphism in the intergenic region (73%) compared to that in the transcribed region, and the lack of missense/nonsense/frame-shift mutations in the exons. This phenomenon is not uncommon. It has been reported, for the human species, that, on average, about 69% of the SNPs are located in intergenic regions versus 31% of the SNPs located in genic regions [38]. Out of them only a small fraction (0.76%) has been reported to be located in the exons and, roughly, half of them have been described as missense mutations, while a 3:1 proportion of the missense to silent mutations would be expected under neutrality (absence of selection) [38]. In the *C. dromedarius MSTN* gene, the proportion of exonic versus genic SNPs was 6:27 (roughly, 0.22%), with no missense or nonsense SNPs observed, possibly reflecting an evolutionary constraint. Indeed, under the harsh desert conditions, more than in other environments, hyper-muscularity may represent a clear metabolic disadvantage, emphasizing thermoregulatory and water availability challenges typical of these habitats. A support to this hypothesis is also offered by the allele frequency patterns observed for the six exonic SNPs detected at the *C. dromedarius MSTN* gene, where the balanced allele frequencies (Table 1) nicely match with the results of the Tajima’s D neutrality test (Appendix A), the latter pointing to a possible balancing selection effect.

### 3.2. Comparative Analysis

A comparative analysis carried out by BLASTing the *C. dromedarius* myostatin sequence (GCA_000803125.3) against the corresponding sequences in *C. ferus* (GCA_009834535.1) and *C. bactrianus* (GCA_000767855.1), for the 99 loci segregating within the *C. dromedarius* species in the region, including the ±5 kbp upstream and downstream of *MSTN*, highlighted that *C. bactrianus* and *C. ferus* displayed the same allele (i.e., the same nucleotide) of the *C. dromedarius* reference sequence at 36 and 48 loci, respectively. Out of them, 31 and 43 loci, respectively, harbored the allele shown to be the most frequent within our *C. dromedarius* considered cohort (consensus allele, in Appendix A). The nucleotide at the G/T SNP locus (position 58,466,681 bps) segregating within *C. dromedarius* displays, in *C. ferus*, a single base deletion. The number of loci where *C. bactrianus* and *C. ferus* share the same allele is 85/99, in line with the results from previous studies on divergence time evaluation within the *Camelus* genus [39,40], thus pointing, as expected, to a genetic distance of *C. dromedarius* from the wild and domestic two-humped camels higher than the genetic distance between *C. bactrianus* and *C. ferus.*

### 3.3. In Silico Functional Prediction

A bioinformatic analysis using the TFBIND tool was conducted to identify whether the allelic variations of the SNPs and indels in the target region resulted in differential binding of any transcription factors (TFs). Overall, in the 2 kbps proximal to the transcription initiation site of the *C. dromedarius* myostatin gene, 11 variants (i.e., nine SNPs and two indels) were identified. By using the two alternative alleles for each variant, a total of 213 differential putative TF-binding sites (TFBS) (out of which 82 were unique) were observed (Appendix A); out of which 41 (14 being unique) relative, overall, to ten TFs with known literature evidence supporting their involvement in muscle metabolism and/or muscle development, i.e., COMP1 (cooperates with myogenic proteins 1, 2), C/EBP (CCAAT/enhancer binding protein, 12), MYOG (Myogenin, 2), COUP (Chicken Ovalbumin Upstream Promoter-transcription factor II, 2), MEF2 (Myocyte Enhancer Factor-2, 16), SRF (Serum Response Factor, 1), DELTAEF1 (Zinc Finger E-Box Binding Homeobox 1, 1), PBX1 (PBX Homeobox 1, 3), MYOD (Myoblast Determination Protein 1, 1) and E2F (Retinoblastoma-Binding Protein, 1). Out of them, MEF2, DELTAEF1 and MYOG are involved in promoting and regulating the skeletal muscle cell differentiation program during myogenesis [41,42,43]. Moreover, MYOG and MYOD belong to a family of proteins known as myogenic regulatory factor (MRF) that plays a major role in regulating the skeletal muscle differentiation [44]. Interestingly, MYOD may also regulate muscle repair [45]. SRF is required for MyoD expression and, consequently, for the myogenic differentiation and maintenance of muscle fibers [46]. A study conducted in *Xenopus laevis* suggested that the RSRFC4 transcription factor may regulate muscle-specific transcription in embryos, and may have other roles during muscle development [47]. Besides, in the skeletal muscles, E2F is essential for the full activation of the myogenic genes during muscle growth and myofibrillogenesis [48], and COMP1 plays a role in the interaction with myogenin [49]. PBX1 is considered a pioneer factor, since it is constitutively bound to the promoter of the myogenin gene, serving as a platform for the MYOD binding in inactive chromatin, thereby preparing the genes of the skeletal muscle lineage for activation [50]. Lastly, COUP-TFII inhibits myogenesis and skeletal muscle metabolism in vitro and in vivo, by repressing the myoblast fusion [51]. Similarly, all of the C/EBP isoforms have a role in the inhibition of myogenic differentiation [52,53].

We next focused our attention on the possible mechanism of alternative splicing; therefore, the HSF tool was used to predict the effects of the identified variant sites on splicing signals. In detail, HSF was designed to perform in silico predictions for the formation or disruption of splice donor sites, splice acceptor sites, branching points (BPs), exonic splicing silencer sites (ESSs) and exonic splicing enhancer sites (ESEs). We performed the mentioned analysis on the variants falling in the exonic, intronic and UTR regions of the gene by separately feeding HSF, for each variant, with the sequences harboring the two alternative alleles. As for the effects on the splice donor and acceptor sites (Table 2), out of the 27 analyzed variants (20 intronic, 6 exonic and 1 in the 5′ UTR), six variants were predicted to disrupt three natural splice sites and to create four alternative splice sites (three acceptor and one donor splice sites).

Concerning the effects on the branching points, eight SNPs were predicted to cause the creation of eight putative branching points and the disruption of three putative BPs (Appendix A). A total of 22 variants were found to alter the ESEs (Appendix A) and 25 variants were found to alter the ESSs (Appendix A). In particular, as to what concerns the variants responsible for the formation/disruption of the splice donor/acceptor sites, we focused our attention on: (i) the single T/TAATAA indel at position 58,464,910 bps, as this was the only indel among those predicted to alter the splice donor/acceptor sites and (ii) the SNP at position 58464775, as this was the only locus to be responsible for the alteration of all of the categories of the splice elements (donor/acceptor sites, branching points, exonic splicing silencer/enhancer sites). Simulating retention of the 793 bps region of intron 1, subsequent to the creation of an alternative donor splice site when the TAATAA allele at position 58,464,910 bps is present, the expected protein would be prematurely truncated due to the presence of a stop codon only two amino acids downstream of the canonical end of the peptide encoded by the exon 1 (Figure 1A). Given the potential relevance of such an observation, we checked whether the TAATAA allele was present in a homozygous state in at least one animal in our cohort, but it appeared that the seven copies of the TAATAA allele were all in a heterozygous state in the seven animals harboring the insertion. These animals were sampled in different countries (Tunisia, Libya, Mauritania, Sudan), with the exception of two samples coming from Morocco but belonging to two different sub-populations (Guerzni and Marmouri). Although all of the animals bearing the TAATAA allele were of African (mainly North-African) origin, still, the geographically scattered distribution of this allele may suggest the presence of homozygous animals as rather uncommon. Similarly to our results, Ceccobelli et al., also in Marchigiana cattle, [54] did not found homozygote-mutated animals in the investigated dataset of 78 samples, and observed only a light superiority in the live weight reached by heterozygous bulls compared to wild-type bulls, although it was not significant. It is indeed known that the incredibly high muscularity typical of the double muscle phenotype generally occurs in association with a homozygous condition for the mutated allele at the myostatin locus. Mosher et al. [16] discovered a 2-bp deletion in the whippet dog *MSTN* gene that, in the homozygote state, results in a double-muscling phenotype commonly referred to as the “bully” whippet, while the heterozygote animals display a phenotype of intermediate musculature. Qian et al. [55] demonstrated that the homozygote *MSTN* mutant pigs had an apparent double muscle phenotype, and individual muscle mass increased by 100% over their wild-type controls at eight months of age as a result of myofiber hyperplasia, while the *MSTN*-mutant heterozygotes animals had much lower muscle mass than the homozygote *MSTN* mutant pigs.

Interestingly, a prematurely truncated protein would also be expected when simulating the retention of the 871-bps region of intron 1, subsequent to the creation of an alternative acceptor splice site when the C allele at position 58,464,775 bps is present instead of the G allele (Figure 1B). A total of 78 animals harbored, in our study cohort, the C allele and, intriguingly, all of them presented a heterozygous genotype. This time, almost all of the countries represented in our 183 sample set had at least one heterozygous animal, with Morocco having the highest number of heterozygous animals (13).

The alternative splicing mechanisms can affect gene expression, thereby generating several distinct proteins [56] and enriching the phenotypic diversity among the individuals in a population [57]. To date, few studies focusing on the on-lab detection of alternative splicing have been conducted in *C. dromedarius,* among which are included the one by Premraj et al. [58] that reported four tissue-specific transcript variants of the interleukin-26 (IL-26) gene; the one by Ryskaliyeva et al. [59] that identified two isoforms of the dromedary αs2-CN milk phosphoprotein/protein arising from differential splicing events, increasing the ability of camel caseins to generate potentially bioactive peptides; and the one by Kappeler et al. [60] that highlighted the presence of two lactophorin variants, resulting from alternative mRNA splicing, in lactating mammary glands of dromedaries. As for the myostatin gene, a previous study from our group [21] did not find any evidence for the alternative splicing in this species.

## 4. Conclusions

This study contributed to obtaining a wider and clearer picture of the sequence variation at the *C. dromedarius* myostatin locus. Previously undescribed polymorphisms (SNPs and indels) were detected, and the variant sites previously published by our group were confirmed in a larger world-wide cohort. The myostatin gene was shown to harbor a low sequence variability in *C. dromedarius*, possibly reflecting a specific evolutionary constraint.

On the other hand, the environmental and/or human-mediated selection toward intermediately high muscular mass, that is more compatible (i.e., less metabolically challenging) with life in extreme xeric habitats compared to extreme, homozygous hyper-muscularity, may be the reason for the observed lack, in our study cohort, of animals homozygous for the alleles in silico predicted to significantly alter the myostatin-splicing process and, consequently, its amino-acidic sequence and function.

Further efforts need to be devoted to collecting reliable phenotypic data on dromedary muscularity and racing performances, thus allowing the unveiling of possible genotype–phenotype associations useful (i) to better understand the molecular bases of muscle-mass development in dromedaries and (ii) to possibly assist selection for meat and athletic traits in this species via the large-scale genotyping of validated genetic markers. Indeed, although in a non-uniform manner, the intensification of dromedary meat production systems is rapidly ongoing in many of the traditional dromedary countries, together with a growing business turnover for the dromedary athletic competition sector. Under the above scenario, it is very likely that genotype-assisted selection may soon also come under the spotlight in this species.

## Figures and Tables

**Figure 1 animals-12-02068-f001:**
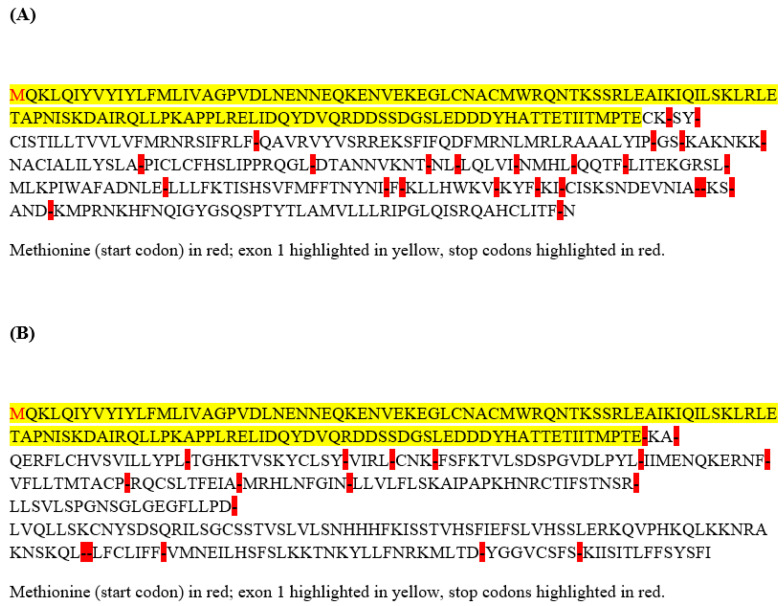
Expected amino acid sequences based on HSF results. (**A**) Simulation of retention of 793 bps in intron 1 due to the creation of an alternative donor splice site when the TAATAA allele belonging to 5var58464910 is present; (**B**) Simulation of retention of 871 bps in intron 1 due to the creation of an alternative acceptor splice site when the C allele belonging to 5var58464775 is present.

**Table 1 animals-12-02068-t001:** Analysis of potential amino acid substitutions caused by SNPs in the coding region.

#	Variant	Project	Chromosome	Allele 1	Allele 2	Base Codon	Mutation	Allele
Exon	Name	ID *	Position **			Position	Type	Frequency
1	5var58465820	PRJEB55295	58465820	T	C	3rd	Silent	0.5
2	5var58463788	PRJEB55295	58463788	G	T	3rd	Silent	0.5
3	5var58461341	PRJEB55295	58461341	G	A	3rd	Silent	0.5
3	5var58461338	PRJEB55295	58461338	A	T	3rd	Silent	0.5
3	5var58461332	PRJEB55295	58461332	T	C	3rd	Silent	0.5
3	5var58461263	PRJEB55295	58461263	A	G	3rd	Silent	0.5

* Accession number for the project submitted to European Variant Archive (EVA, https://www.ebi.ac.uk/eva/ accessed on 7 August 2022). ** The position refers to the CamDro3 assembly (GCA_000803125.3).

**Table 2 animals-12-02068-t002:** Differential acceptor and donor splice sites identified by the HSF sequence analysis using the two sequences harboring the alternative alleles.

	ALLELE 1			ALLELE 2		
**INTRONIC** **VARIANTS**	**Type ^1^**	**Motif ^2^**	**Value ^3^**	**Type ^1^**	**Motif ^2^**	**Value ^3^**
5var58464953	-	-	-	Acceptor splice site	CCCGGTCTGCAGAT	81.96
5var58464910	Donor splice site	TATGTTATT	72.09	-	-	-
5var58464775	Acceptor splice site	GCACCTTAACAGAG	77.89	-	-	-
5var58464463	-	-	-	Donor splice site	GGTGTTAAT	66.52
5var58464142	Donor splice site	GAAGTAGGT	81.61	Acceptor splice site	GAAAGGAAGCAGGT	68.36
**EXONIC VARIANTS**	**Type ^1^**	**Motif ^2^**	**Value ^3^**	**Type ^1^**	**Motif ^2^**	**Value ^3^**
5var58461332	-	-	-	Acceptor splice site	ATTGCACCTAAGAG	74.03

^1^ Type of splicing signals: acceptor splice sites or donor splice sites; ^2^ Sequence motif harboring the predicted splicing signals; ^3^ The signals values range from 65 (weak) to 100 (very strong).

## Data Availability

The variant data for this study have been deposited in the European Variation Archive (EVA) at EMBL-EBI under accession number PRJEB55295 (https://www.ebi.ac.uk/eva/?eva-study=PRJEB55295).

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
