# Peer review of "Refining the Camelus dromedarius Myostatin Gene Polymorphism through Worldwide Whole-Genome Sequencing"

_animals, 2022, doi:10.3390/ani12162068_

Round 1

Reviewer 1 Report

Overall, this is well written manuscript describing the variation in the Myostatin gene in Camelids. It presents a large body of information generated using Whole Genome Next Gen Sequencing. The analysis and presentation are good. The conclusions are appropriate.

 Some minor comments:

Table 1. Of the 6 snps in the exonic regions did any of these result in possible rare codon useage?

Line 210 “It has been reported, for the human species, that, on average, about 69% of SNPs are located in inter-genic regions versus 31% of SNPs located in genic regions.” I presume this is from reference 33?

Line 330: rather than using “novel” I would use “previously undescribed”.

Line 334: With this available sequence information – have you considered looking to see if there are any regions in the C. dromedarius Myostatin protein displaying purifying selection?

Reviewer 2 Report

Dear authors,

although the theme of myostatin is not so new, it was not considered in the Camelidae family. However, this article is showing only MSTN locus polymorphism and the prediction of their effect, without translating to phenotype. Nevertheless, I think that this article is interesting for some readers. The manuscript is well written. Below are minor comments:

line 82-92. - It will be curious to add information if the meat and milk, and wool of these camels are also used in other non-Arabian countries?

Material and methods: did your sequencing data publicly available? if yes please add the accession number.

Table 1 - amino acid column is not necessary here should be indicated that these mutations are silent.

results and discussion

I understood that in silico analysis concerned the predictions. Did the authors attempt to compare camel phenotype vs. MSTN variants, especially those which are associated with changes in TF binding or donor-acceptor splice sides? particularly that some of indicated TFs are related to muscle development.

line 266-276 is any information in the literature on alternative splicing in camels?

the conclusion surprised me a little. There was pinpointed that selection toward intermediately high muscle mass is more compatible with life in extreme xeric habitats. It means that selection forward to higher muscle mass would not be an advantage for the camels, not in accordance with their welfare? So the potential MSTN markers are not needed? I think that the conclusion should be a little rearranged. 

Reviewer 3 Report

Dear author,

Thank you for submitting the paper entitled "Refining the Camelus dromedarius myostatin gene polymor- 2 phism through worldwide whole-genome sequencing". Although the setting is simple, I found the reading very interesting and the methodology very clear and effective. The work demonstrates how much work remains to be done following the intense sequencing work, especially in marginal species such as the camel. The text is clear and the discussion convincing. Here are just a few observations that I found important for the correctness of the manuscript.

Line 146: Rstudio is only the graphic interface to R. Please specify if you used some specific package for dataset management (e.g. Tidyverse?) and please specify the version of both library used and R integrated in R studied

Line 147: please give more and clearer detail about the range used. Searching in NBCI the reported reference sequence, the MSTN gene is about 6700 pb so you take a total of ~16.000 bp (6700+5000down+5000up)?

Table 1, variant name: one suggestion would be to submit relevant (or all) founded variation to GenBank and report instead the accession number, this could allow in future to other researcher to specifically refer to your finding and helping the increase of information about this gene in the specie. Also, even if I understand that the name of the variant is a pasting or chromosome, and position would be more elegant and effective report the specific position in a dedicated column
